

# Risk and the Point of No Return for Climate Action

Matthias Aengenheyster[1], Qing Yi Feng[2], Frederik  van der Ploeg[3], and Henk
A.  Dijkstra[2]

[1]Atmospheric, Oceanic and Planetary Physics, Department of Physics, Oxford University, Oxford,
UK
[2]Institute for Marine and Atmospheric Research Utrecht, Department of Physics, Utrecht
University, Utrecht, the Netherlands
[3]Centre for the Analysis of Resource Rich Economies, Department of Economics, Oxford
University, Oxford, UK

*Correspondence to:* H.A. Dijkstra (H.A.Dijkstra@uu.nl)

**Abstract.** If the Paris targets are to be met, there may be very few years left for policy makers
to start cutting emissions. Here, we ask by what year at the latest one has to take action to keep
global warming below the $2\,\mathrm{K}$ target (relative to preindustrial levels) at the year 2100 with a 67%
probability; we call this the Point of No Return (PNR). Using a novel, stochastic model of $CO_2$
concentration and global mean surface temperature derived from the CMIP5 ensemble simulations,
we find that cumulative $CO_2$ emissions from 2015 onwards may not exceed $424\,\mathrm{GtC}$ and that the
PNR is 2035 for the policy scenario where the share of renewable energy rises by 2% per year.
Pushing this increase to 5% per year delays the PNR until 2045. For the $1.5\,\mathrm{K}$ target, the carbon
budget is only $198\,\mathrm{GtC}$ and there is no time left before starting to increase the renewable share by
2% per year. If the risk tolerance is tightened to 5%, the PNR is brought forward to 2022 for the
$2\,\mathrm{K}$ target and has been passed already for the $1.5\,\mathrm{K}$ target. Including substantial negative emissions
towards the end of the century delays the PNR from 2035 to 2042 for the $2\,\mathrm{K}$ and to 2026 for the
$1.5\,\mathrm{K}$ target, respectively. We thus show the impact on the PNR not only of the temperature target
and the speed by which emissions are cut, but also of risk tolerance, climate uncertainties and the
potential for negative emissions.

## 1  Introduction

The Earth System is currently in a state of rapid warming that is unprecedented even in geological
records (Pachauri et al., 2014). This change is primarily driven by the rapid increase in atmospheric
concentrations of greenhouse gases (GHG) due to anthropogenic emissions since the industrial rev-
olution (Myhre et al., 2013). Changes in natural physical and biological systems are already being
observed (Rosenzweig et al., 2008), and efforts are made to determine the 'anthropogenic impact'
on particular (extreme weather) events (Haustein et al., 2016). Nowadays, the question is not so
much if, but by how much and how quickly the climate will change as a result of human interfer-



ence, whether this change will be smooth or bumpy (Lenton et al., 2008) and whether it will lead to
dangerous anthropogenic interference with the climate (Mann, 2009).

The climate system is characterized by positive feedbacks causing instabilities, chaos and stochastic dynamics (Dijkstra, 2013) and many details of the processes determining the future behavior of the climate state are unknown. The debate on action on climate change is therefore focused on the question of *risk* and how the *probability* of dangerous climate change can be reduced. In scientific
and political discussions, targets on 'allowable' warming (in terms of change in Global Mean Surface Temperature (GMST) relative to pre-industrial conditions[1]) have turned out to be salient, with the $2\,\mathrm{K}$ warming threshold commonly seen as a safe threshold to avoid the worst effects that might occur when positive feedbacks are unleashed (Pachauri et al., 2014). Indeed, in the Paris COP21 conference it was agreed to attempt to limit warming below $1.5\,\mathrm{K}$ (United Nations, 2015). It is,
however, questionable whether the commitments made by countries (the so-called Nationally Determined Contributions (NDCs)) are sufficient to keep temperatures below the 1.5 K and possibly even the 2.0 K target (Rogelj et al., 2016a).

A range of studies has appeared to provide insight on the safe level of cumulative emissions to stay below either the $1.5\,\mathrm{K}$ or $2.0\,\mathrm{K}$ target at a certain time in the future with a specified probability,
usually taken as the year 2100. Early studies made use of Earth System Models of Intermeditate Complexity (EMICs) (Zickfeld et al., 2009; Huntingford et al., 2012; Steinacher et al., 2013) to obtain such estimates. Because it was found that peak warming depends on cumulative carbon emissions $E_\Sigma$ but is independent of the emission pathway (Allen et al., 2009; Zickfeld et al., 2012), focus has been on the specification of a safe level of $E_\Sigma$ values corresponding to a certain temper-
ature target. In more recent papers, also emulators derived from either C4MIP models (Benjamin M Sanderson, 2016) or CMIP5 (Coupled Model Intercomparison Project 5) models (Millar et al., 2017b), with specified emission scenarios, were used for this purpose. Such methodology was recently used in (Millar et al., 2017a) to argue that a post-2015 value of $E_\Sigma \sim 200$ GtC would limit post-2015 warming to less than $0.6°$C (so meeting the 1.5 K target) with a probability of 66%.

In this paper we pose the following question: assume one wants to limit warming to a specific threshold in the year 2100, while accepting a certain risk tolerance of exceeding it, then, when, at the latest, does one have to start to ambitiously reduce fossil fuel emissions? The point in time when it is 'too late' to act in order to stay below the prescribed threshold is called (van Zalinge et al., 2017) the Point of No Return (PNR). The value of the PNR will depend on a number of quantities, such as the
climate sensitivity and the means available to reduce emissions. To determine estimates of the PNR, a model is required of global climate development that a) is accurate enough to give a realistic picture of the behavior of GMST under a wide range of climate change scenarios, b) is forced by fossil fuel emissions, c) is simple enough to be evaluated for a very large number of different emission and mitigation scenarios and d) provides information about risk, i.e., it cannot be purely deterministic.

---

[1]We define pre-industrial temperature as the 1861-1880 mean temperature, in accordance with IPPC AR5.





The models used in van Zalinge et al. (2017) are clearly too idealized to determine adequate estimates of the PNR under different conditions. In this paper, we therefore construct a stochastic model from the CMIP5 results where many global climate models were subjected to the same forcing for a number of climate change scenarios (Taylor et al., 2012). This stochastic model is then used together with a broad range of mitigation scenarios to determine estimates of the PNR under different

risk tolerances.

  If the Paris temperature targets are to be met, only a few years are left for policy makers to take action by cutting emissions (Stocker, 2013): with an emissions reduction rate of $5\,\%\,\mathrm{yr}^{-1}$, the $1.5\,\mathrm{K}$ target has become unachievable and the $2.0\,\mathrm{K}$ target becomes unachievable after 2017. The Stocker (2013) analysis highlights the crucial concept of the closing door or PNR of climate policy, but it is

deterministic. It does not take account of the possibility that these targets are not met, and does not allow for negative emissions scenarios. We here show how the considerable climate uncertainties captured by our stochastic state-space model of the carbon dynamics and temperature inertia, the degree to which policy makers are willing to take risk, and the potential of negative emissions affect the carbon budget and the date at which climate policy becomes unachievable (the PNR). The climate

policy is here not defined as an exponential emission reduction as in Stocker (2013) but as a steady increase in the share of renewable energy in total energy.

## 2 Methods

We let $\Delta T$ be the annual-mean area-weighted Global Mean Surface Temperature (GMST) deviation from pre-industrial conditions of which the 1861-1880 mean is considered to be representative

(Pachauri et al., 2014; Schurer et al., 2017). From the CMIP5 scenarios we use the simulations of the pre-industrial control, abrupt quadrupling of atmospheric $CO_2$, smooth increase of 1% $CO_2$ per year, and the RCP (Representative Concentration Pathways) scenarios 2.6, 4.5, 6.0 and 8.5 (Taylor et al., 2012). The data is obtained from the German Climate Computing Center (DKRZ), the ESGF Node at DKRZ, and KNMI's Climate Explorer. The $CO_2$ forcings (concentrations (Meinshausen

et al., 2011) and emissions (van Vuuren et al., 2007; Clarke et al., 2007; Fujino et al., 2006; Riahi et al., 2007)) are obtained from the RCP Database (available at http://tntcat.iiasa.ac.at/RcpDb).

  As all CMIP5 models are designed to represent similar (physical) processes but use different formulations, parametrizations, resolutions and implementations, the results from different models offer a glimpse into the (statistical) properties of future climate change, including various forms of

uncertainty. We perceive each model simulation as one possible, equally likely, realization of climate change. Applying ideas and methods from statistical physics (Ragone et al., 2016), in particular Linear Response Theory (LRT), a stochastic model is constructed that represents the CMIP5 ensemble statistics of the GMST.



### 2.1 Linear Response Theory

We use only those ensemble members from CMIP5 for which the control run and at least one perturbation run are available, leading to 34 members for the abrupt ($CO_2$ quadrupling) and 39 for the smooth-forcing experiment. Considering those members from the RCP runs also available in the abrupt forcing run, we have 25 members for RCP2.6, 30 for RCP4.5, 19 for RCP6.0 and 29 for RCP8.5.

The $CO_2$ concentration as a function of time for the abrupt quadrupling and smooth $CO_2$ increase is prescribed as

$$C_{CO2,abrupt}(t) = C_0(3\theta(t) + 1) \tag{1}$$

$$C_{CO2,smooth}(t) = \begin{cases} C_0 & , \quad t \leq 0 \\ C_0 1.01^t & , \quad t > 0 \end{cases} \tag{2}$$

with time in years from the start of the forcing, pre-industrial $CO_2$ concentration $C_0$ and Heaviside
function $\theta(t)$. The radiative forcing $\Delta F$ due to $CO_2$ relative to pre-industrial conditions is given as

$$\Delta F = \alpha_{CO2} \ln\left(\frac{C_{CO2}(t)}{C_0}\right) \tag{3}$$

with $\alpha_{CO2} = 5.35\,\mathrm{W\,m^{-2}}$ (Myhre et al., 2013). With LRT, the Green's function for the temperature response is computed from the abrupt forcing case as the time derivative of the mean response (Ragone et al., 2016)

$$G_T(t) = \frac{1}{\Delta F_{abrupt}} \frac{d}{dt} \Delta T_{abrupt} \tag{4}$$

where $\Delta F_{abrupt}(t) = \ln(4C_0/C_0) = \ln(4)$. The temperature deviation from the pre-industrial state for any forcing $\Delta F_{any}$ in then obtained, via the convolution of the Green's function, as

$$\Delta T_{any}(t) = \int_0^t G_T(t')\Delta F_{any}(t - t')\,dt' \tag{5}$$

Because equation (4) is exact we expect that (5) with $\Delta F_{any} = \Delta F_{abrupt}$ will exactly reproduce
the abrupt CMIP5 response. In addition, for the LRT to be a useful approximation, the response has to reasonably reproduce the smooth $1\,\%\,\mathrm{yr}^{-1}$ CMIP5 response with $\Delta F_{any} = \Delta F_{smooth}$. Figure 1a shows that LRT applied to the abrupt perturbation recovers perfectly – as required – the abrupt response and is well able to recover the response to a smooth forcing. The correspondence is very good for the mean response and also the variance is captured quite well. In order to apply LRT
to the RCP scenarios, the radiative forcing has to be scaled up by a constant factor $A$ as these - unlike the idealized abrupt and smooth scenarios - include non-fossil $CO_2$ emissions and non-$CO_2$ GHG emissions. The constant $A = 1.48$ was found in order to optimize the agreement of $\Delta T$ with CMIP5. The resulting reconstruction of temperatures from RCP $CO_2$ concentrations overlaid with CMIP5 data (Figure 1b), also gives a good agreement.



Beyond finding the temperature change as a result of $CO_2$ variations, eventually emissions $E_{CO2}$
cause these $CO_2$ changes and have to be addressed explicitly. A multi-model study of many carbon
models of varying complexity under different background states and forcing scenarios was recently
presented (Joos et al., 2013). A fit of a three-timescale exponential with constant offset was proposed
for the ensemble mean of responses to a $100\,\mathrm{GtC}$ emission pulse to a present-day climate of the form


$$G_{CO2}(t) = a_0 + \sum_{i=1}^{3} a_i e^{-\frac{t}{\tau_i}} \tag{6}$$

Coefficients $a_i, i = 0 \ldots 3$ and timescales $\tau_i, i = 1 \ldots 3$ are determined using least-square fits on the
multi-model mean. The $CO_2$ concentration then follows from

$$C_{CO2}(t) = \int_0^t G_{CO2}(t')\, E_{CO2}(t - t')\, dt' \tag{7}$$

In doing so, we use a response function that is independent of the size of the impulse, i.e. the carbon
cycle reacts in the same way to pulses of all sizes other than $100\,\mathrm{GtC}$. This is of course a simpli-
fication, especially as very large pulses might unleash positive feedbacks to do with the saturation
of natural sinks such as the oceans (Millar et al., 2017b), but works reasonably well in the range of
emissions we are primarily interested in.

The full (temperature and carbon) LRT model is summarized as

$$
\begin{align}
C_{CO2}(t) &= C_{CO2,0} + \int_0^t G_{CO2}(t')\, E_{CO2}(t - t')\, dt' \tag{8a}\\
\Delta F_{CO2}(t) &= A\, \alpha_{CO2} \ln(C_{CO2}(t)/C_0) \tag{8b}\\
\Delta T(t) &= \Delta T_0 + \int_0^t G_T(t')\Delta F_{CO2}(t - t')dt' \tag{8c}
\end{align}
$$

and relates fossil $CO_2$ emissions $E_{CO2}$ to mean GMST perturbation $\Delta T$ with initial conditions
$C_{CO2,0}$ for $CO_2$ and $\Delta T_0$ for GMST perturbation. This is quite a simple model with few 'knobs
to turn'. The only really free parameter is the constant $A$ that scales up $CO_2$-radiative forcing to
take into account non-fossil $CO_2$ and non-$CO_2$ GHG emissions. Internally, emissions need to be
converted from $\mathrm{GtC\,yr^{-1}}$ to $\mathrm{ppm\,yr^{-1}}$ using the respective molar masses and the mass of the Earth's
atmosphere as $E_{CO_2}[\mathrm{ppm\,yr^{-1}}] = \gamma E_{CO_2}[\mathrm{GtC\,yr^{-1}}]$ with $\gamma = 0.469\,69\,\mathrm{ppm\,GtC^{-1}}$. In Table 1 we
summarize our estimates of the model's ten parameters.

In Figure 2 we show the results obtained for RCP emissions. For very high emission scenarios we
underestimate $CO_2$ concentrations because for such emissions natural sinks saturate. However, the
up-scaling of radiative forcing is quite successful, yielding a good temperature reconstruction.



### 2.2 Stochastic State Space Model

The model outlined above still contains a data-based temperature response function and it informs only about the *mean* CMIP5 response. However, our main motivation is to obtain new insights on the possible evolution to a 'safe' carbon-free, state and such paths necessarily depend strongly on the variance of the climate and on the risk one is willing to take. This variance in temperature is quite substantial, as is evident from Figure 1b. Therefore we translate our response function model to a state-space model and incorporate the variance via suitable stochastic terms.

The response function $G_T$ from the 140-year abrupt quadrupling ensemble is well approximated by

$$G_T(t) = \sum_{i=0}^{2} b_i e^{-\frac{t}{\tau_{bi}}} \tag{9}$$

Although $\tau_{b0} \to \infty$, we require a finite $\tau_{b0}$ for temperatures to stabilize at some level. Hence, we choose a long time scale $\tau_{b0} = 400\,\mathrm{yr}$ that cannot really be determined from the $140\,\mathrm{yr}$ abrupt forcing (CMIP5) runs. By writing

$$C = C_P + \sum_{i=1}^{3} C_i \tag{10a}$$

$$\Delta T = \sum_{i=0}^{2} \Delta T_i \tag{10b}$$

the LRT model can be transformed into the 7-dimensional Stochastic State Space Model (SSSM) shown in Table 2 with parameters in Table 3. Initial conditions are obtained by running the noise-free model forward from pre-industrial conditions ($C_P = C_0$ and $C_i = \Delta T_i = 0, i = 1, 2, 3$) to present-day, driven by historical emissions [2]. As these temperatures are now given relative to the start of emissions, i.e. 1765, we add the 1961-1990 model mean to the HadCRUT4 dataset to get observed temperature deviation relative to 1765, and compute $\Delta T$ relative to 1861-1880 by adding the 1861-1880 mean of this deviation time series.

The major benefit of this formulation is that we can include stochasticity. We introduce additive noise to the carbon model such that the standard deviation of the model response to an emission pulse as reported by (Joos et al., 2013) is recovered. For the temperature model we introduce (small) additive noise to recover the (small) CMIP5 control run standard deviation. In the CMIP5 RCP runs the ensemble variance increases with rising ensemble mean. This calls for the introduction of (substantial) multiplicative noise, which we introduce in $\Delta T_2$, letting these random fluctuations decay over an 8-year timescale. The magnitude of these fluctuations is (especially at high temperatures) likely to be unrealistic when looking at individual time series. However, the focus here is on ensemble statistics.

---

[2]these are the fossil fuel and cement production emissions from (Le Quéré et al., 2016), accessed $28^{th}$ March, 2017



### 2.3 Transition Pathways

The SSSM described in the previous section is forced with fossil $CO_2$ emissions. We assume that, in the absence of any mitigation actions, emissions increase from their initial value $E_0$ at an exponential rate $g$ due to economic and population growth. Political decisions cause emissions to decrease from starting year $t_s$ onward as fossil energy generation is replaced by non-GHG producing forms such as wind, solar and water (mitigation $m$) and by an increasing share of fossil energy sources the emissions of which are not released but captured and stored away by Carbon Capture and Storage (abatement $m$). In addition, negative emission technologies $E_{neg}$ may be employed that lead to a net reduction in atmospheric $CO_2$ concentration. We model this in a very simple way by letting both mitigation and abatement increase linearly until emissions are brought to zero:

$$m(t) = \begin{cases} m_0 & t \leq t_s \\ \min(m_0 + m_1(t - t_s), 1) & t > t_s \end{cases} \tag{11a}$$

$$a(t) = \begin{cases} a_0 & t \leq t_s \\ \min(a_0 + m_1(t - t_s), 1) & t > t_s \end{cases} \tag{11b}$$

$$E(t) = E_0 e^{gt}(1 - a(t))(1 - m(t)) - E_{neg}(t) \tag{11c}$$

with constants $m_0, a_0$ giving the mitigation and abatement rates at the start of the scenario and $m_1$ the incremental year-to-year increase. The simplified model (11) is very well able (not shown) to reproduce the IAM pathways from that fulfil the NDCs until 2030 and afterwards reach the 2 K target with a 50-66% probability (Rogelj et al., 2016a). These pathways are exemplary for those that continue on the low-commitment path for a while, followed by strong and decisive action.

### 2.4 Point of No Return

With the emission scenarios and the SSSM - returning $CO_2$ concentrations and GMST for any such scenario - one can now address the issue of transitioning from the present-day (year 2015) to a carbon-free era such as to avoid catastrophic climate change. We need to take into account both the *target* threshold and the *risk* one is willing to take to exceed it. The maximum amount of cumulative $CO_2$ emissions that allows reaching the 1.5 and 2 K targets, as a function of the risk tolerance, is called the Safe Carbon Budget (SCB). It is well established in the literature (Meinshausen et al., 2009; Zickfeld et al., 2009) but does not contain information on how these emissions are spread in time. This is where the Point of No Return (PNR) comes in: The PNR is the point in time where starting mitigating action is insufficient to stay below a specified target with a chosen risk tolerance.

Concretely, let the temperature target $\Delta T_{max}$ be the maximum allowable warming and denote the parameter $\beta$ as the probability of staying below a given target (a measure of the risk tolerance). For example the case $\Delta T_{max} = 2\,\text{K}$ and $\beta = 0.9$ corresponds to a 90% probability of staying below $2\,\text{K}$



warming, i.e. 90 of 100 realizations of the SSSM, started in 2015 and integrated until 2100, do not exceed $2\,\mathrm{K}$ in the year 2100.

Then, in the context of (11), the PNR is the earliest $t_s$ that does not result in reaching the defined 'Safe State' (van Zalinge et al., 2017) in terms of $\Delta T_{max}$ and $\beta$. It is determined from the probability distribution $p(\Delta T_{2100})$ of GMST in 2100. Both SCB and PNR depend on temperature target, climate uncertainties and risk tolerance, but the PNR also depends on the aggressiveness of the climate action considered feasible (here given by the value of $m_1$). This makes the PNR such an interesting quantity, since the SCB does not depend on the time path of emission reductions. Clearly there is a close connection between the PNR and the SCB. Indeed, one could define a PNR also in terms of the ability to reach the SCB. The one-to-one relation between cumulative emissions and warming gives the PNR in 'carbon space'. Its location in time, however, depends crucially on how fast a transition to a carbon-neutral economy is feasible.

Since it is now recognized that negative emissions may be essential in meeting temperature targets, we include this possibility into the PNR computation. From the IAM scenarios that Rogelj et al. (2016a) found to fulfill NDCs until 2030 and stay below $2\,\mathrm{K}$ with 50-66% probability, we obtain a family of negative emission pathways (Figure 3) out of which we pick a 'moderate' (orange) and a 'strong' (red) pathway.

## 3   Results

To demonstrate the quality of the SSSM we initialise it at pre-industrial conditions, run it forward and compare the results with those of CMIP5 models. The SSSM is well able to reproduce the CMIP5 model behavior under the different RCP scenarios (Figure 4, shown for RCP2.6 and 4.5). As these scenarios are very different in terms of rate of change and total cumulative emissions this is not a trivial finding. It is actually remarkable that the SSSM, which is based on a limited amount of CMIP5 model ensemble members, performs so well. As an example, the RCP2.6 scenario contains substantial negative emissions, responsible for the downward trend in GMST, which our SSSM correctly reproduces. The mean response for RCP8.5 is slightly underestimated (not shown) because the uncertainty in the carbon cycle plays a rather minor role compared to that in the temperature model. In addition, for such large emission reductions positive feedback loops set in from which our SSSM abstracts. The temperature perturbation $\Delta T$ is very closely log-normally distributed while for weak forcing scenarios (e.g., RCP2.6 and RCP4.5) the distribution is approximately Gaussian. The $CO_2$ concentration is found to be Gaussian distributed for all RCP scenarios. These findings (log-normal temperature and Gaussian $CO_2$ concentration) result from the multiplicative and additive noise in temperature and carbon components of the SSSM, respectively.

To determine the SCB, 6000 emission reduction strategies (with $E_{neg}(t) = 0$) were generated and, using the SSSM, an 8000-member ensemble for each of these emission scenarios starting in 2015





was integrated. Emission scenarios are generated from (11) by letting $a(t) = 0$, a uniform $m_0 \in [0, 0.7]$ and $m_1$ drawn from a beta distribution (with distribution function $p(m) = \frac{1}{B(\alpha, \delta)} m^\alpha (1 - m)^{(\delta-1)}$, where $B(\alpha, \delta)$ is the beta function; parameters are chosen as $\alpha = 1.2, \delta = 3$), with the

[0,1] interval scaled such that $m = 1$ latest in 2080.

The temperature anomaly in 2100 ($\Delta T_{2100}$) as a function of cumulative $CO_2$ emissions $E_\Sigma$ is shown in Figure 5. The same calculation is also shown for the deterministic case without climate uncertainty (no noise in the SSSM). In Figure 5, the SCB is given by the point on the $E_\Sigma$-axis where the (colored) line corresponding to a chosen risk tolerance crosses the (horizontal) line corresponding

to a chosen temperature threshold $\Delta T_{max}$. The curves $\Delta T_{2100} = f(E_\Sigma)$ (Figure 5) are very well described by expressions of the type

$$f(E_\Sigma) = a \ln \left( \frac{E_\Sigma}{b} + 1 \right) + c \tag{12}$$

with suitable coefficients $a, b$ and $c$, each depending on the tolerance $\beta$. For the range of emissions considered here, a linear fit would be reasonable (Allen et al., 2009). However, our expression also

works for cumulative emissions in the range of business as usual (when fitting parameters on suitable emission trajectories). From Figure 5 we easily find the SCB for any combination of $\Delta T_{max}$ and $\beta$, as shown in Table 4.

Allowable emissions are drastically reduced when enforcing the target with a higher probability (following the horizontal lines from right to left in Figure 5). These results show in particular the

challenges posed by the $1.5\,K$ compared to the $2\,K$ target. The sensitivity of the SCB to the relevant model parameters is shown in the Appendix and the values are robust. From IPCC-AR5 (IPPC, 2013) we find cumulative emissions post-2015 of $377\,GtC$ to $517\,GtC$ in order to 'likely' stay below $2\,K$ while we find an SCB of $424\,GtC$ for $\Delta T_{max} = 2\,K, \beta = 0.67$ which lies in the same range. Like Millar et al. (2017a) we find approximately $200\,GtC$ to stay below $2\,K$ with $\beta = 0.67$.

To determine the PNR, we resort to three illustrative choices to model the abatement and mitigation rates with $E_{neg}(t) = 0$. Following (11) we construct Fast Mitigation (FM) and Moderate Mitigation (MM) scenarios with $m_1 = 0.05$ and $0.02$, respectively. In addition, in an Extreme Mitigation (EM) scenario $m = 1$ can be reached instantaneously. This corresponds to the most extreme physically possible scenario and serves as an upper bound. When varying $t_s$ to find the PNR for

the three scenarios, we always keep $m_0 = 0.14$ and $a_0 = 0$ at 2015 values (World Energy Council, 2016).

As an example, $t_s = 2025$ leads to total cumulative emissions from 2015 onward of 109, 183 and $335\,GtC$ for the mitigation scenarios EM, FM and MM, respectively. Note that while MM is the most modest scenario, it is actually quite ambitious, considering that with $m = 0.1355$ in 2005 and

$m = 0.14$ in 2015 (World Energy Council, 2016) the current year-to-year increases in the share of renewable energies are very small.

Figure 6 shows the probabilities for staying below the 1.5 and $2.0\,K$ thresholds in 2100 as function of $t_s$ for different policies, including FM ($m_1 = 0.05$) and MM ($m_1 = 0.02$), while the EM policy





bounds the unachievable region. It is clear that this region is larger for the 1.5 than for the 2.0

degree target, and shrinks when including negative emissions. From the plot we can directly see the
consequences of delaying action until a given year. For example, if policy makers should choose to
implement the MM strategy only in 2040, the chances of reaching the 1.5 (2.0) degree target are
only 2% (47%). We conclude that the remaining 'window of action' may be small, but a window
still exists for both targets. For example, the $2\,\mathrm{K}$ target is reached with a probability of 67% even

when starting MM is delayed until 2035. However, reaching the $1.5\,\mathrm{K}$ target appears unlikely as
MM would be required to start in 2018 for a probability of 67%. When requiring a high ($\geq 0.9$)
probability, it is impossible to reach with the MM scenario. The PNR for the different targets and
probabilities is given in Table 5. The robustness of these PNR values is shown in the Appendix.

We also see from Figure 6 and Table 7 that the inclusion of negative emissions delays the PNR by

6-10 years (see Table 7), which may be very valuable especially for ambitious targets. For example,
when including 'strong' negative emissions one can reach $1.5\,\mathrm{K}$ with a probability of up to 66% in
the MM scenario when acting before 2026, 8 years later than without. The PNR varies substantially
for slightly different temperature targets. This also illustrates the importance of the temperature
baseline relative to which $\Delta T$ is defined. This has been found previously (Schurer et al., 2017), and

we find (not shown) that switching to an $18^{th}$ century baseline can move the PNR earlier by up to
10 years.

It is clear that an energy transition more ambitious than RCP2.6 is required to stay below $1.5\,\mathrm{K}$
with some acceptable probability, and whether that is feasible is doubtful. For all other RCP scenarios, exceeding $2\,\mathrm{K}$ is very likely in this century (Figure 7).

300    ## 4   Summary, Discussion and Conclusions

We have developed a novel stochastic state space model (SSSM) to accurately capture the basic
statistical properties (mean and variance) of the CMIP5 RCP ensemble, allowing us to study warming probabilities as function of emissions. It represents an alternative to the approach that contains
stochasticity in the parameters rather than the state. Although the model is highly idealized, it cap-

305    tures simulations of both temperature and carbon responses to RCP emission scenarios quite well.

A weakness of the SSSM is the simulation of temperature trajectories beyond 2100 and for high
emission scenarios. The large multiplicative noise factor leads – especially at high mean warmings –
to immensely volatile trajectories that in all likelihood are not physical (on the individual level, the
distribution is still well-behaved). It might be a worthy endeavour to investigate how this could be

310    improved. Another weakness in the carbon component part of the SSSM is that the real carbon cycle
is not pulse-independent. Hence, using a single constant response function has inherent problems, in
particular when running very high-emission scenarios. This is because the efficiency of the natural
carbon sinks to the ocean and land reservoirs is a function both of temperature and the reservoir



sizes. The SSSM has therefore slight problems reproducing $CO_2$ concentration pathways (Figure 2), a price we accept to pay as we focus on the CMIP5 temperature reproduction. Taking account of non-$CO_2$ emissions more fully beyond our simple scaling and avoiding temporary overshoots of the temperature caps would reduce the carbon budgets (Rogelj et al., 2016b) and thus lead to earlier PNRs than given here. Therefore the values might be a little too optimistic.

In Millar et al. (2017b), the authors draw a different conclusion from studying a similar problem. They introduce in their FAIR model response functions that dynamically adjust parameters based on warming to represent sink saturation. Consequently, their model gives much better results in terms of $CO_2$ concentrations. It would be an interesting lead for future research to conduct our analysis here (in terms of SCB and PNR) with other simple models (such as FAIR or MAGICC) to discover similarities and differences. However, only rather low-emission scenarios are consistent with the 1.5 or 2 K targets, so we do not expect this to play a major role, and indeed our carbon budgets are very similar to Millar et al. (2017a).

The concept of a Point of No Return introduces a novel perspective into the discussion of carbon budgets that is often centered on the question of when the remaining budget will have 'run out' at current emissions. In contrast, the PNR concept recognizes the fact that emissions will not stay constant and can decay faster or slower depending on political decisions. With these caveats in mind, we conclude that, first, the PNR is still relatively far away for the 2.0 K target: with the MM scenario and $\beta = 67\%$ we have 17 years left to start. When allowing to set all emissions to zero instantaneously, the PNR is even delayed to the 2050s. Considering the slow speed of large-scale political and economic transformations, decisive action is still warranted, as the MM scenario is a large change compared to current rates. Second, the PNR is very close or passed for the 1.5 K target. Here more radical action is required – 9 years remain to start the FM policy to avoid 1.5 K with a 67% chance, and strong negative emissions gives us 8 years under the MM policy.

Third, we can clearly show the effects of changing $\Delta T_{max}, \beta$ and the mitigation scenario. Switching from 1.5 to 2 K buys an additional $\approx 16$ years. Allowing a one-third, instead of one-tenth exceedance risk, buys an additional 7-9 years. Allowing for the more aggressive FM policy instead of MM buys an additional 10 years. This allows to assess trade-offs, for example between tolerating higher exceedance risks and implementing more radical policies. Fourth, negative emissions can offer a brief respite but only delay the PNR by a few years, not taking into account the possible decrease in effectiveness of these measures in the long term (Tokarska and Zickfeld, 2015).

We have shown the constraints put on future emissions by restricting GMST increase below 1.5 and 2 K, respectively, and the crucial importance of the safety probability. Further (scientific and political) debate is essential on what are the right values for both temperature threshold and probability. Our findings are sobering in light of the bold ambition in the Paris agreement, and add to the sense of urgency to act quickly before the PNR has been crossed.



*Acknowledgements.* We thank the focus area 'Foundations of Complex Systems' of Utrecht University for
providing the finances for the visit of F. van der Ploeg to Utrecht in 2016. HAD acknowledges support by
the Netherlands Earth System Science Centre (NESSC), financially supported by the Ministry of Education,
Culture and Science (OCW), Grant no. 024.002.001.



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





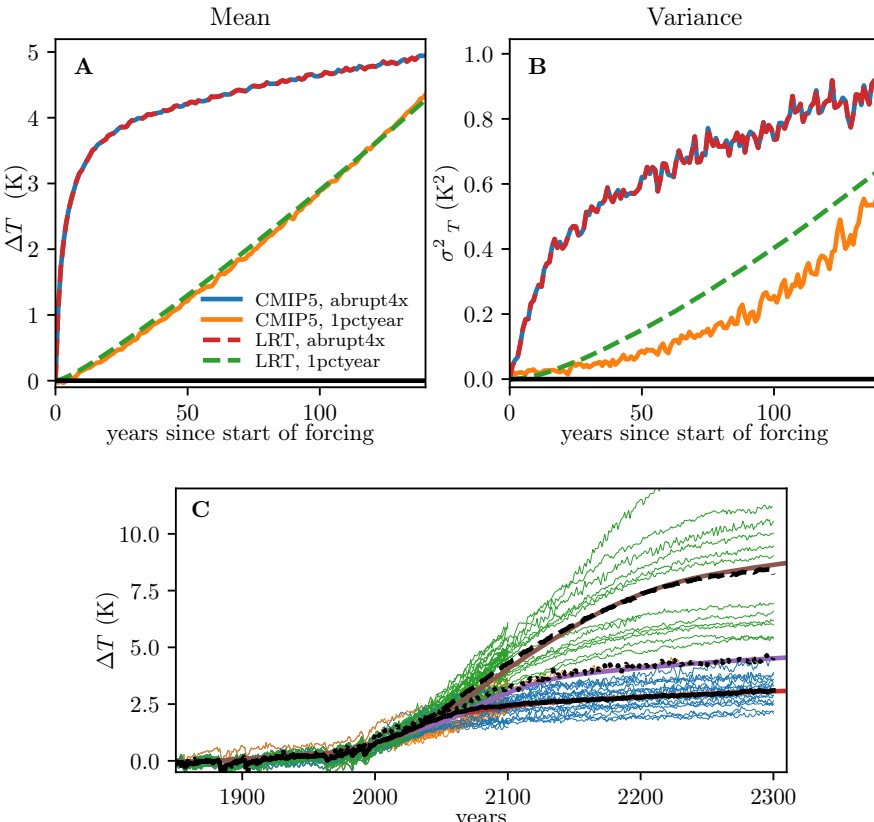

**Figure 1.** Ensemble mean (**A**) and variance (**B**) of temperature response from CMIP5 (solid) and LRT reproduction (dashed). Year 0 gives the start of the perturbation. (**C**) Reconstruction of RCP temperature evolution from concentration pathways using $CO_2$ only. Blue, orange and green lines gives CMIP5 data for RCP4.5, RCP6.0 and RCP8.5, respectively, with the ensemble mean given in black solid (RCP4.5), dotted (RCP6.0) and dashed (RCP8.5) black. Reconstruction using $CO_2$ radiative forcing in red (RCP4.5), purple (RCP6.0) and brown (RCP8.5).



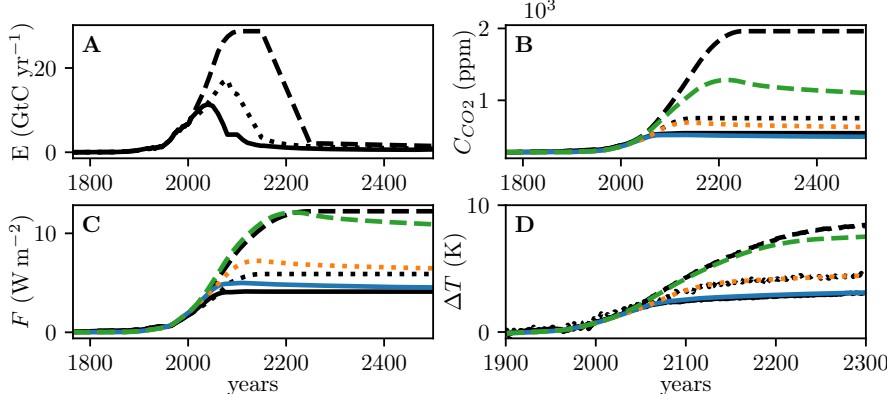

**Figure 2.** Reconstruction of RCP results using the Response Function Model. In all panels, solid lines refer to RCP4.5, dotted to RCP6.0 and dashed lines to RCP8.5. Black lines show RCP data while colors (blue: RCP4.5, orange: RCP6.0, green: RCP8.5) give our reconstruction. **(A):** Fossil $CO_2$ emissions. **(B):** $CO_2$ concentrations from RCP and reconstructed using $G_{CO2}$. **(C):** Total anthropogenic radiative forcing (black) and radiative forcing from $CO_2$ only (red) (both from RCP) and reconstructed forcing using the relations above. **(D):** Temperature perturbation from CMIP5 RCP (ensemble mean) and the our reconstruction.





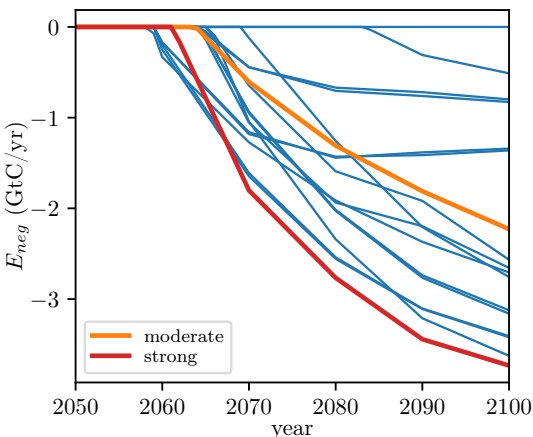

**Figure 3.** Negative Emissions from IAM scenarios (Rogelj et al., 2016a), with two sample pathways marked.

| $C_0$ (ppm) | $a_0$ | $a_1$ | $a_2$ | $a_3$ |
|---|---|---|---|---|
| 278 | 0.2173 | 0.2240 | 0.2824 | 0.2763 |
| $A$ | $\alpha$ $(\mathrm{W\,m^{-2}})$ | $\tau_1$ | $\tau_2$ | $\tau_3$ |
| 1.48 | 5.35 | 394.4 | 36.54 | 4.304 |

**Table 1.** Response Function Model Parameters. All timescales $\tau_i$ are in years and the carbon model amplitudes $a_i$ are dimensionless for $E$ in $\mathrm{ppm\,yr^{-1}}$.





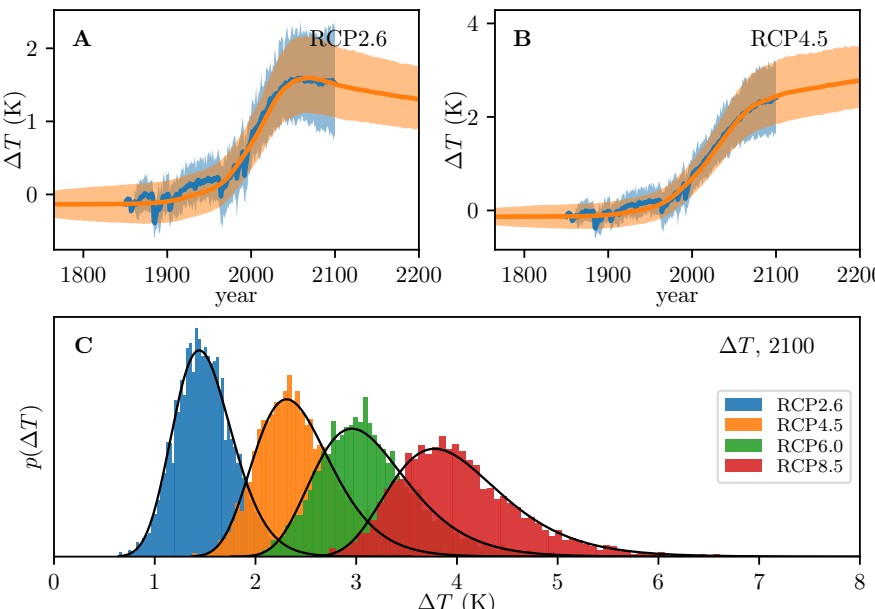

**Figure 4.** Stochastic State Space Model applied to RCP scenarios. **(A,B):** Ensemble mean and $5^{th}$, $95^{th}$ percentile envelopes of CMIP5 RCPs (blue) and stochastic model (orange). **(C):** Probability density functions for $\Delta T$ in 2100 based on 5000 ensemble members, and driven by forcing from RCP2.6 (blue), RCP4.5 (orange), RCP6.0 (green) and RCP8.5 (red). In black are fitted lognormal distributions.




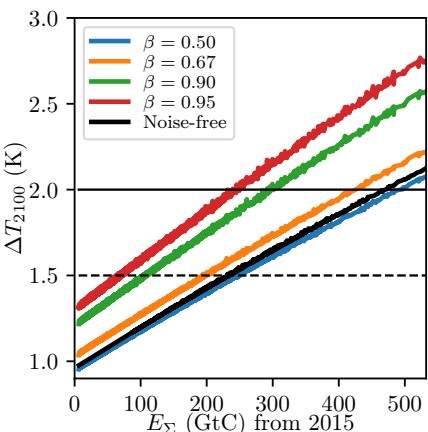

**Figure 5.** The Safe Carbon Budget. $\Delta T_{max}$ in 2100 such that $p(\Delta T_{2100} \leq \Delta T_{max}) = \beta$ as a function of cumulative emissions for different $\beta$. The black curve gives the deterministic results with noise terms in the stochastic model set to zero.



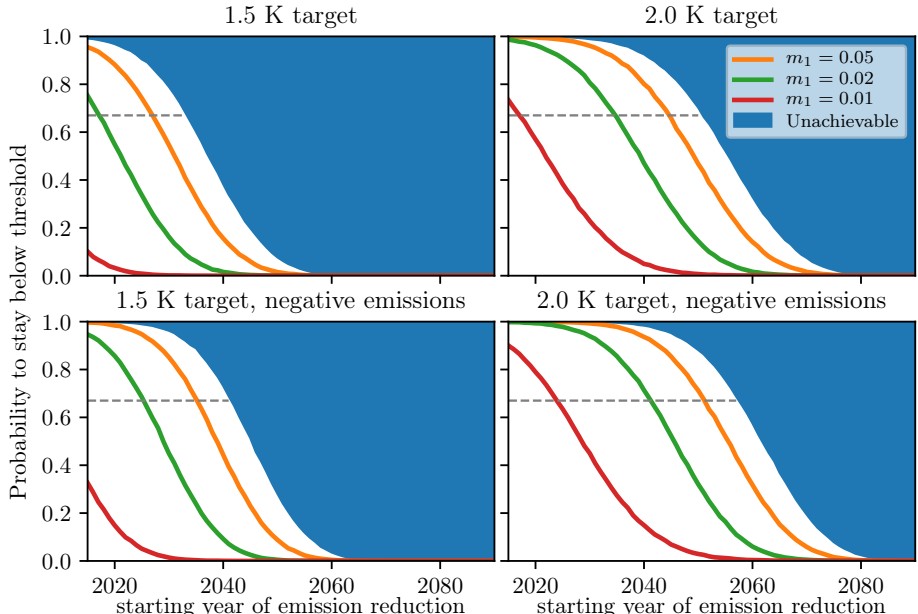

**Figure 6.** The Point of No Return. Probability of staying below the 1.5 K (left) or 2.0 K (right) threshold when starting emission reductions in a given year, for different policies, without (top) and with (bottom) strong negative emissions. The Point of No Return for a given policy is given by the point in time where the probability drops below a chosen threshold. The default threshold of two-thirds is dashed. The unachievable region is bounded by the extreme mitigation scenario.

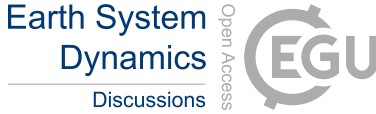



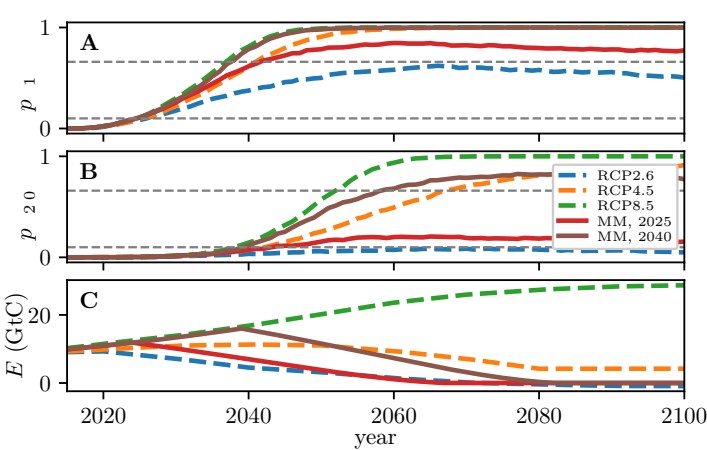

**Figure 7. (A,B):**Instantaneous probability to exceed 1.5 K **(A)** and 2.0 K **(B)** for different emission scenarios. RCP scenarios are shown as dashed lines while solid lines give MM scenario results starting in 2025 (red) and 2040 (brown). Dashed horizontal lines give $p = 0.1$ and 0.67, respectively. **(C)**: Fossil fuel emissions in GtC for the same scenarios.



$$dC_P = a_0 E dt$$

$$dC_1 = (a_1 E - \frac{1}{\tau_1} C_1) dt$$

$$dC_2 = (a_2 E - \frac{1}{\tau_2} C_2) dt + \sigma_{C2}\, dW_t$$

$$dC_3 = (a_3 E - \frac{1}{\tau_3} C_3) dt$$

$$C = C_P + \sum_{i=1}^{3} C_i$$

$$\Delta F = A\, \alpha \ln(C/C_0)$$

$$d\Delta T_0 = (b_0 \Delta F - \frac{1}{\tau_{b0}} \Delta T_0) dt + \sigma_{T0}\, dW_t$$

$$d\Delta T_1 = (b_1 \Delta F - \frac{1}{\tau_{b1}} \Delta T_1) dt$$

$$d\Delta T_2 = (b_2 \Delta F - \frac{1}{\tau_{b2}} \Delta T_2) dt \quad + \sigma_{T2} \Delta T_2\, dW_t$$

$$\Delta T = \sum_{i=0}^{2} \Delta T_i$$

**Table 2.** Stochastic State Space Model. Carbon model on left, temperature model on the right.

| $a_0$ | $a_1$ | $a_2$ | $a_3$ | $\tau_1$ | $\tau_2$ | $\tau_3$ |
|---|---|---|---|---|---|---|
| 0.2173 | 0.2240 | 0.2824 | 0.2763 | 394.4 | 36.54 | 4.304 |

| $C_0$ (ppm) | $b_0$ | $b_1$ | $b_2$ | $\tau_{b0}$ | $\tau_{b1}$ |
|---|---|---|---|---|---|
| 278 | 0.00115176 | 0.10967972 | 0.03361102 | 400 | 1.42706247 |

| $A$ | $\alpha$ (W m$^{-2}$) | $\sigma_{C2}$ (ppm/yr$^{1/2}$) | $\sigma_{T0}$ (K/yr$^{1/2}$) | $\sigma_{T2}$ (yr$^{-1/2}$) | $\tau_{b2}$ |
|---|---|---|---|---|---|
| 1.48 | 5.35 | 0.65 | 0.015 | 0.13 | 8.02118539 |

**Table 3.** Stochastic State Space Model Parameters. All timescales are in years, the carbon model amplitudes $a_i$ are dimensionless for $E$ in ppm yr$^{-1}$, the temperature model amplitudes $b_i$ are in K W$^{-1}$ m$^2$ yr$^{-1}$.





| $\beta$ | 0.5 | 0.67 | 0.9 | 0.95 | Noise-free |
|---|---|---|---|---|---|
| $T_{max} = 1.5\,\mathrm{K}$ | 247 | 198 | 107 | 69 | 233 |
| $T_{max} = 2.0\,\mathrm{K}$ | 492 | 424 | 298 | 245 | 469 |

**Table 4.** Safe Carbon Budget (in GtC since 2015) as function of threshold and safety probability $\beta$.

| | $\beta$ | 0.5 | 0.67 | 0.9 | 0.95 | noise-free |
|---|---|---|---|---|---|---|
| EM | $T_{max} = 1.5\,\mathrm{K}$ | 2038 | 2034 | 2026 | 2022 | 2037 |
| | $T_{max} = 2.0\,\mathrm{K}$ | 2056 | 2051 | 2042 | 2038 | 2055 |
| FM | $T_{max} = 1.5\,\mathrm{K}$ | 2032 | 2027 | 2020 | 2016 | 2030 |
| | $T_{max} = 2.0\,\mathrm{K}$ | 2050 | 2045 | 2036 | 2032 | 2048 |
| MM | $T_{max} = 1.5\,\mathrm{K}$ | 2022 | 2018 | – | – | 2021 |
| | $T_{max} = 2.0\,\mathrm{K}$ | 2040 | 2035 | 2026 | 2022 | 2038 |

**Table 5.** Point of No Return as function of threshold and safety probability $\beta$ without negative emissions.

| | $\beta$ | 0.5 | 0.67 | 0.9 | 0.95 | noise-free |
|---|---|---|---|---|---|---|
| EM | $T_{max} = 1.5\,\mathrm{K}$ | 2046 | 2042 | 2035 | 2032 | 2045 |
| | $T_{max} = 2.0\,\mathrm{K}$ | 2062 | 2058 | 2049 | 2046 | 2061 |
| FM | $T_{max} = 1.5\,\mathrm{K}$ | 2039 | 2036 | 2028 | 2025 | 2038 |
| | $T_{max} = 2.0\,\mathrm{K}$ | 2056 | 2052 | 2043 | 2039 | 2055 |
| MM | $T_{max} = 1.5\,\mathrm{K}$ | 2029 | 2026 | 2019 | – | 2029 |
| | $T_{max} = 2.0\,\mathrm{K}$ | 2046 | 2042 | 2033 | 2030 | 2045 |

**Table 6.** Point of No Return as function of threshold and safety probability $\beta$ with strong negative emissions.

| | $\beta$ | 0.5 | 0.67 | 0.9 | 0.95 | no-noise |
|---|---|---|---|---|---|---|
| EM | $T_{max} = 1.5\,\mathrm{K}$ | 8 | 8 | 9 | 10 | 8 |
| | $T_{max} = 2.0\,\mathrm{K}$ | 6 | 7 | 7 | 8 | 6 |
| FM | $T_{max} = 1.5\,\mathrm{K}$ | 7 | 9 | 8 | 9 | 8 |
| | $T_{max} = 2.0\,\mathrm{K}$ | 6 | 7 | 7 | 7 | 7 |
| MM | $T_{max} = 1.5\,\mathrm{K}$ | 7 | 8 | (4) | – | 8 |
| | $T_{max} = 2.0\,\mathrm{K}$ | 6 | 7 | 7 | 8 | 7 |

**Table 7.** Difference of PNR between strong and no negative emissions. Values in parentheses if no PNR exists without negative emissions, PNR is then assumed to have been 2015.



**Appendix: SCB and PNR Parameter Sensitivity**

SCB and PNR sensitivities were determined by varying each parameter by $\pm10\%$ and running the calculation to see how the obtained value changes. Sensitivities were determined for all discussed

values of $T_{max}, \beta$, and the EM, FM and MM scenarios in case of PNR. We show (Table 8) sample values for a small ($T_{max} = 1.5\,\mathrm{K}, \beta = 0.95$), intermediate ($T_{max} = 1.5\,\mathrm{K}, \beta = 0.5$), and large ($T_{max} = 2.0\,\mathrm{K}, \beta = 0.5$) SCB, corresponding to a close, intermediate and far PNR.

The biggest effects on the SCB are found for the initial condition of the large carbon reservoirs and the radiative forcing parameters $A, \alpha$ and $C_0$ that are essentially fixed constants. The parameters

of the carbon model ($a_i, \tau_i$) do not have big impacts on the found SCB, on the order of $0 - 17\,\mathrm{GtC}$, with the larger numbers found for larger absolute values of SCB. Varying the temperature-model parameters can have quite noticeable effects, up to 10% for large and up to 50% for small values of SCB. The model is particularly sensitive to changes in the intermediate timescale ($b_2, \tau_{b2}$). Likely, possible variations in the (model) parameters are not independent, potentially canceling each other.

The sensitivity of SCB and PNR to the noise amplitudes is small, with largest values found for the multiplicative noise amplitude that is responsible for much of the spread of the temperature distribution (so increasing $\sigma_{T2}$ decreases the SCB).

The PNR sensitivities are generally small and in no way change our message qualitatively. The effect of initial conditions and carbon model parameters is small, often even unnoticeable (with the

exception of the permanent carbon reservoir, due to its large size). We find the most relevant, yet small, sensitivities in the temperature model parameters. For example, a 10% error in $\tau_{b2}$ can move the PNR by 2-3 years. An interesting effect is the case of $r_\gamma$, the energy-saving progress (reduction in energy-intensity of a unit of economic output and in effect equivalent to a decrease in the emission growth rate) which is taken zero by default. Increasing it to 1% or 2% has little effect on *close* PONR

(e.g. 2020) but is capable of delaying *late* PNR by up to 15 years, and the effect is more substantial for the less ambitious scenarios. This is an interesting finding, showing that in the long run increasing energy efficiency can play a role in avoiding the PNR.





| $T_{max}, \beta$ | SCB | | | PNR | | |
|---|---|---|---|---|---|---|
| | $1.5K, 0.95$ | $1.5K, 0.5$ | $2.0K, 0.5$ | $1.5K, 0.95$ | $1.5K, 0.5$ | $2.0K, 0.5$ |
| undisturbed | 68.63 | 247.02 | 492.09 | 2022 | 2036 | 2050 |
| $C_1$ | 15.06, -15.01 | 14.75, -14.48 | 14.65, -13.41 | 2, -1 | 1, -1 | 1, 0 |
| $C_2$ | 1.59, -2.07 | 1.96, -2.0 | 1.47, -1.88 | 0, 0 | 0, 0 | 0, 0 |
| $\Delta T_1$ | -0.12, -0.05 | 0.09, -0.0 | 0.52, 0.04 | 0, 0 | 0, 0 | 0, 0 |
| $\Delta T_2$ | -0.04, -0.03 | 0.05, 0.1 | -0.04, -0.49 | 1, 0 | 0, 0 | 0, 0 |
| $a_1$ | 2.81, -2.82 | 10.24, -9.41 | 19.52, -17.2 | 0, 0 | 0, -1 | 1, -1 |
| $a_2$ | 0.68, -0.79 | 3.27, -2.91 | 7.76, -6.49 | 1, 1 | 0, -1 | 1, 0 |
| $\tau_1$ | 3.64, -3.02 | 4.73, -3.75 | 5.92, -4.43 | 0, 0 | 0, 0 | 1, 0 |
| $\tau_2$ | 4.58, -4.48 | 7.6, -7.1 | 12.44, -11.08 | 1, 0 | 0, -1 | 1, 0 |
| $A$ | 55.59, -44.99 | 80.98, -64.43 | 118.57, -93.23 | 5, -4 | 5, -5 | 6, -5 |
| $\alpha$ | 55.76, -44.97 | 80.91, -64.52 | 118.18, -92.85 | 5, -4 | 5, -5 | 6, -5 |
| $C_0$ | –, 169.67 | -188.37, 182.48 | -205.7, 199.12 | –, 13 | -15, 11 | -12, 10 |
| $b_1$ | 12.17, -11.57 | 22.74, -21.04 | 32.55, -31.19 | 1, -1 | 2, -2 | 2, -1 |
| $b_2$ | 32.08, -28.29 | 38.94, -34.41 | 57.89, -50.29 | 4, -2 | 2, -3 | 3, -2 |
| $\tau_{b1}$ | 12.31, -11.83 | 23.02, -21.17 | 34.51, -30.64 | 1, -1 | 1, -2 | 2, -1 |
| $\tau_{b2}$ | 37.84, -33.21 | 38.13, -33.51 | 56.77, -49.36 | 4, -3 | 2, -3 | 3, -2 |
| $\gamma_0$ | $\sim, \sim$ | $\sim, \sim$ | $\sim, \sim$ | 1, -1 | 2, -2 | 3, -2 |
| $r_\gamma$ | $\sim, \sim$ | $\sim, \sim$ | $\sim, \sim$ | 1, 1 | 2, 5 | 6, 15 |
| $\sigma_{T2}$ | 10.04, -10.16 | -3.0, 3.55 | -4.68, 5.15 | 1, -1 | 0, 0 | 0, 0 |

**Table 8.** Sensitivity of Safe Carbon Budget and Point of No Return to parameter variations. Values as difference in $\mathrm{GtC}$ (SCB) and number of years (PNR) from the undisturbed value (first row). The PNR values all refer to the EM scenario. First and second numbers give $10\%$ parameter decrease and increase, respectively. Exception is $r_\gamma$ (in orange) which is zero by default and where first and second numbers give $r_\gamma = 0.01$ and $r_\gamma = 0.02$, respectively. No sensitivities are calculated for the SCB for the economic parameters $\gamma_0$ and $r_\gamma$ and replaced by ($\sim$), whereas (–) implies no positive SCB/PNR could be calculated. The fields corresponding to the radiative forcing parameters $A, \alpha, C_0$ are colored in cyan, while the most sensitive climate model parameters $b_2, \tau_{b2}$ are given in orange.