# Peer review of "The Point of No Return for climate action: effects of climate uncertainty and risk tolerance"

_Earth System Dynamics, 2018_

## Referee Comment (RC1) · Anonymous Referee #1 · 25 Mar 2018

This paper provides a definition of a point of no return for avoiding specified temperature increases with a given likelihood. The authors develop a stochastic model to address this issue based on $CO_2$ concentration and temperature simulated in the CMIP5 ensemble runs. The paper addresses the sensitivity of the point of no return to a number of factors and with a set of assumptions about rates of decarbonisation of the economy. The method and results are clear and the assumptions are well declared. I see no major issues with this paper and provide only minor comments below:

1. The authors assess temperature rises to 2100 only. For some scenarios the global mean temperature will continue to increase well after this date. Those cases should be acknowledged.

2. The authors note that 2K warming is commonly seen as a "safe threshold". It may

be seen that way, but that is a value judgment subject to considerable uncertainty, and this should be acknowledged.

3. The assessment of delta T depends on the baseline period chosen. This point is addressed later in the report and is said to introduce a sensitivity to the PNR of up to 10 years. The new IPCC special report on warming of 1.5C and 2C indicates potentially large differences in delta T for different baseline choices. It would be nice to see the authors address this issue more explicitly to have confidence that their PNR sensitivity is as low as reported.

4. The authors use the concept of "negative emissions" in their simulations, but don't say much about the feasibility of negative emissions. Some elaboration would be helpful for the reader.

5. The trajectory of warming from the present point to exceeding the specified temperature threshold will not be smooth as it will include multidecadal scale internal variability. That implies that the threshold will not be exceeded at a single point in time, but only in some average sense. The degree to which this is an issue depends on how well the CMIP5 runs represent multidecadal internal variability and how one treats temporal variability and overshoot in relation to the threshold. The authors could provide some discussion of this issue in relation to their analysis.

---

## Referee Comment (RC2) · Anonymous Referee #2 · 26 Apr 2018

The authors present a very simple climate-carbon cycle emulator that is tuned to CMIP5 ensemble simulations. They address the question of timing of the "point of no return" (PNR), i.e. the time when a certain mitigation policy will not longer guarantee that the global mean temperature increase relative to preindustrial stays below a given temperature target, e.g. 1.5 or 2°C with a certain probability. The simplicity of the model enables the authors to carry out large ensembles from which they can determine the PNR under different probabilities and targets. The authors venture to give precise years for PNR. This may actually be misleading given the host of uncertainties that are associated with such approaches, and mitigation projections in general. In principle, this is a valuable study that could eventually be published. However, I raise a few concerns and questions below, that the authors need to address, before I can

[Figure]

recommend publication.

Comments:

1. The new approach is essentially twofold: first a very simple deterministic model is developed that reproduces global characteristics of CMIP5, and second, emission pathways are given as an exponential increase at rate g (information not found in the paper: g=??) multiplied by a linearly decreasing factor (mitigation effect). In addition, negative emissions due to carbon capture and storage can be considered in this model framework. It would be useful to quantify the difference of the considered paths (11c) to an even more basic choice of just a simple exponential decrease of emissions at a constant rate from t_s onwards, as used by Stocker (2013). Obviously, the discontinuity of emission rates at t_s (increasing exponentially before, and then decreasing) are avoided here, but how would that matter for the PNR? Incidentally, for a given mitigation rate PNR can be read off Fig. 2A of Stocker (2013): it is the required starting time of emission reductions. Therefore, much of the information, which is the focus of the present paper, has been available already from an even simpler framework. This should be mentioned in the introduction.

2. Uncertainty is only substantively addressed in the text of the appendix. As this is a short text, I suggest to incorporate the appendix into the main text and amplify it. Regarding uncertainty, a general caveat would be useful in the abstract and the conclusion. Otherwise, the stated years of PNR are somewhat misleading.

3. A constant factor A in the forcing (8b) is used to optimize the agreement with CMIP5. The size of this factor is quite large (1.48). For $\alpha_{CO_2}$ (in 8b) the correct value is taken (see Tab 3 - however inconsistent parameter notation - only $\alpha$ there!). The authors justify the factor A with the existence of non-CO2 GHG drivers in the CMIP5 results (RCP scenarios), but the effects of these drivers have a time evolution and characteristic time scales that are very different from the primary driver CO2. So I don't quite understand how is it possible to achieve a better match with CMIP5 by using

a simple scaling of (8b).

4. It is not clear, why in (11) both mitigation and abatement are used. Also, there is a conflict of parameters ($a_0$ in 6 and 11). Is 11b, i.e. $a(t)$, really needed and relevant in this paper? I see no discussion in the text or the figures relating to the difference of $m(t)$ and $a(t)$ pathways. In fact, inspecting (11c) I can see no benefit why one would consider both mitigation and abatement. Both have the same linear time dependence, even the same rate. Therefore, the difference seems to lien in the quadratic (positive) contribution $a(t)*m(t)$ to the emission factor, essentially $(m1^2)(t-t\_s)^2$, presumably a rather small contribution. Therefore, for simplicity, I suggest that you would eliminate $a(t)$ altogether, which would also remove the parameter conflict of $a_0$.

5. Further to the emission pathway described in 11c, I note that $E\_neg$ is included. However, it is not clear from the text, how Fig. 3 is constructed. From the rather short caption I surmise that this is taken from Rogelj et al., and then just prescribed here. This must be stated in section 2.3 more clearly.

6. You seem to consider only the strong negative emission of Fig. 3 for the calculation of PNR in Tab. 6. As this strong case appears nearly exponential in nature, I would suggest that you simply approximate the Rogelj negative emissions by an exponential and a starting time, and give it explicitly in eq 11 with its associated rate. This would eliminate Fig. 3, be more transparent for the reader and actually more consistent with the simple scenario approach that you chose in eq 11.

7. In order to construct ensembles, the mitigation rate $m\_1$ is drawn from a Beta distribution. It would be helpful for the reader to have an explanation why this distribution is chosen and what difference a simple uniform or normal distribution would make.

8. Some noise is added to the model as stated on line 167ff. It seems of only minor relevance for the results (see Tab 5 and 6 - PNR changes only by about 1 year compared to the 50%-probability case). I wonder then why the addition of noise should be necessary at all. I cannot see any new insight from this. If you retain the noise, a more

detailed description would be necessary. In particular, the noise should be evident in eqs 10a and 10b as additional terms.

9. Table 5, 6, and 7 could be presented in a more effective way. Table 7 is trivial (just the difference Tab6 - Tab5) and could therefore be omitted. I further suggest to combine Tables 5 and 6 into one table. Each probability column should then contain two subcolumns, one without E_neg the other one with E_neg. The small difference caused by E_neg makes would then be directly visible.

10. In the appendix and in Tab. 8 some parameters (\gamma_0, r_\gamma) are listed without explanation. Where do they come from? Are they needed in this paper?

11. Line 374: please spell IPCC correctly. It is an edited document and that information is missing, as well as the total page number.

12. Figure 2: Put the 10ˆ3 factor into the label unit (1000 ppm).

13. Figure 2 and line 147. The discrepancy with the CMIP5 CO2 concentrations for RCP8.5 is quite worrying. This would imply that cumulative emissions will be way off, as well. The discrepancy for the forcing is removed by introducing the factor A, but what about CO2(t) and cumE(t)?? This must be addressed in a more convincing way.

14. Figure 6: Caption should be amplified by elaborating on the "different policies". You could add, e.g.: "... as described by m in eq 11, the rate of mitigation increase per year."

15. Figure 7: y-axis labels not complete.

---

## Author Comment (AC1) · 30 May 2018

Response to Anonymous Referee RC 1

We thank the referee for the careful reading and the useful comments and will adapt the manuscript accordingly. Below is a point by point reply with the referee's comments in bold font, our reply in italic font and the changes in manuscript in normal font.

1. Comment from referee:
   **The authors assess temperature rises to 2100 only. For some scenarios the global mean temperature will continue to increase well after this date. Those cases should be acknowledged.**

   *Author's response:*
   *Such possibility can indeed occur and is implicitly acknowledged in Figures 1, 2 and 4. We do not treat these cases in much detail because a) the response function model is based on 140-year long simulations so extrapolations far into the future are more uncertain and b) such scenarios exceed our temperature targets and are therefore of limited interest in this study.*

   Changes in Manuscript:
   In line 40, after "usually taken as the year 2100." we will add the following sentence: "The choice of a particular year is necessarily arbitrary and neglects the possibility of additional future warming."

2. Comment from referee:
   **The authors note that 2K warming is commonly seen as a "safe threshold". It may be seen that way, but that is a value judgment subject to considerable uncertainty, and this should be acknowledged.**

   *Author's response:*
   *We agree with the referee.*

   Changes in Manuscript:
   In line 32, after "the 2 K warming threshold commonly seen" we will add " – while gauging the considerable uncertainty – ".

3. Comment from referee:
   **The assessment of delta T depends on the baseline period chosen. This point is addressed later in the report and is said to introduce a sensitivity to the PNR of up to 10 years. The new IPCC special report on warming of 1.5C and 2C indicates potentially large differences in delta T for different baseline choices. It would be nice to see the authors address this issue more explicitly to have confidence that their PNR sensitivity is as low as reported.**

   *Author's response:*
   *Within the scope of our model the effect of the baseline on the PNR is such that a lower baseline increases the currently realized warming. Therefore, a given temperature threshold is crossed at an earlier point in time.*

   Changes in Manuscript:
   To clarify this point, the paragraph referring to the temperature baseline (from line 293, "This also") will be replaced by the following: "This also illustrates the importance of the temperature baseline relative to which ΔT is defined, as has been found previously (Schurer et al., 2017). Switching to a (lower) 18[th] century baseline increases current levels of warming by 0.13 K (Schurer et al., 2017) and thereby brings forward the PNR. For example, for a maximum temperature threshold of 1.5K the PNR is brought forward from 2022 to 2016 in the MM scenario and from 2038 to 2033 for the EM scenario."

4. Comment from referee:
   **The authors use the concept of "negative emissions" in their simulations, but don't say much about the feasibility of negative emissions. Some elaboration would be helpful for the reader.**

   *Author's response:*
   *It is not within the scope of this article to provide a detailed discussion of the question of feasibility of negative emissions, which is a research area in its own right. Scenarios such as the ones presented here and taken from Rogelj et al., (2016a) are usually based on cost-minimization in Integrated Assessment Models (IAMs), and are feasible within the constraints and choices enforced there.*

   Changes to Manuscript:
   In line 223, at the end of the paragraph, we will add the sentence "For details on the scenarios refer to Rogelj et al., (2016a). With carbon budgets rapidly running out and the PNR approaching fast,

negative emissions accomplished by geoengineering may have to become an essential part of the policy mix. Such policies are cheap but may only be a temporary fix and lead to undesirable spillover effects on neighboring countries (e.g., Wagner and Weitzman, 2015). We abstract from these discussions here, since this is beyond the scope of the present paper".

We will add: "Wagner, G. and M.L. Weitzman (2015). *Climate Shock. The Economic Consequences of a Hotter Planet*, Princeton University Press, Princeton, New Jersey" to reference list.

5. Comment from referee:
   The trajectory of warming from the present point to exceeding the specified temperature threshold will not be smooth as it will include multidecadal scale internal variability. That implies that the threshold will not be exceeded at a single point in time, but only in some average sense. The degree to which this is an issue depends on how well the CMIP5 runs represent multidecadal internal variability and how one treats temporal variability and overshoot in relation to the threshold. The authors could provide some discussion of this issue in relation to their analysis.

   *Author's response:*
   *It is indeed the case that, due to internal variability, crossing the threshold takes place in some average sense. Commonly this is done by temporal averaging over 30 years. In our case, averaging is done in across the ensemble of simulations. Therefore, it is indeed possible to pinpoint the crossing of the threshold (at a chosen probability level) to a given year, as the large ensemble smooths out the variability (Figure 6). The model is not capable of accurately displaying modes of internal variability, nor is it designed to predict (in a one time-series sense) the crossing of the threshold.*

   Changes to Manuscript:

   Before the final paragraph starting in line 345 ("We have shown the constraints..."), we will add the following paragraph:

   "In this work a large ensemble of simulations was used in order to average over stochastic internal variability. This allows to pinpoint the point in time where a threshold is crossed at a chosen probability level. Such an ensemble is not possible for more realistic models, nor do GCMs agree on details of internal variability. Therefore, in practice, the crossing of a threshold will likely be determined with hindsight and using 30-year temporal means. This fact should lead us to be more cautious in choosing mitigation pathways."

---

## Author Comment (AC2) · 30 May 2018

We thank the referee for the careful reading and the useful comments and suggestions and will adapt the manuscript accordingly. Below is a point by point reply with the referee's comments in bold font, our reply in italic font and the changes in manuscript in normal font.

**General remarks to the referee:**

The referee remarks that giving precise years for the Point of No Return (PNR) may be misleading due to many uncertainties associated with such approaches. This is certainly true and in fact the primary motivation to conduct this study in a probabilistic fashion, with the aim to capture climate system uncertainties in the model itself.

We see the presentation of a stochastic model as a major novelty of this paper, building upon and extending previous work such as Stocker (2013). The aim was to a) include uncertainties as captured by the CMIP5 ensemble and b) get a handle on risk tolerance, allowing us to choose with which probability a certain warming target should not be exceeded. Clearly, tighter constraints (i.e. an earlier Point of No Return (PNR)) are intuitively expected for a smaller risk tolerance but the model allows us to quantify this.

The stochastic state space model is described in section 2.2 and summarized in Table 2 (where also the noise terms are detailed), as stated in line 161. Noise is included in several of the carbon and temperature boxes, where $W_t$ denotes the Wiener process. These boxes are added to form the total $CO_2$ concentration and temperature anomaly (eqs 10a, 10b). The introduction of additive and multiplicative noise is central to this paper, and turns the temperature evolution $DT(t)$ into the evolution of a probability density $p(DT,t)$ (Figure 4), capturing the spread of the CMIP5 ensemble.

**Reponses to the referee's specific comments:**

1. Comment from referee:
   **The new approach is essentially twofold: first a very simple deterministic model is developed that reproduces global characteristics of CMIP5, and second, emission pathways are given as an exponential increase at rate g (information not found in the paper: g=??) multiplied by a linearly decreasing factor (mitigation effect). In addition, negative emissions due to carbon capture and storage can be considered in this model framework. It would be useful to quantify the difference of the considered paths (11c) to an even more basic choice of just a simple exponential decrease of emissions at a constant rate from t_s onwards, as used by Stocker (2013). Obviously, the discontinuity of emission rates at t_s (increasing exponentially before, and then decreasing) are avoided here, but how would that matter for the PNR? Incidentally, for a given mitigation rate PNR can be read off Fig. 2A of Stocker (2013): it is the required starting time of emission reductions. Therefore, much of the information, which is the focus of the present paper, has been available already from an even simpler framework. This should be mentioned in the introduction.**

   *Author's response:*
   - *The referee is right to point out that the original response function model (eqs 8) is deterministic. However, as pointed out in the introductory paragraph, this deterministic model is turned into a stochastic one through the introduction of stochastic noise terms (see section 2.2, Table 2, Figure 4), and used throughout the paper.*
   - *We thank the referee for noticing the omitted definition of the emissions growth rate. It will be corrected.*
   - *In the final paragraph of the introduction (lines 66-76) we refer to Stocker (2013) and point out how our approach differs from his, in particular by using a stochastic model that is capable of capturing climate uncertainties and risk tolerance. We agree with the referee that a comparison of our mitigation pathways (11) with exponential pathways (Stocker) is interesting. Therefore we have performed such an analysis and show the results here, in the manner of Fig. 2A of Stocker (2013). From Figure RC1 one can see that the notable novelty of this work is the introduction of*

[Figure]

*Figure RC1: Reconstruction of Fig. 2A from Stocker (2013) (top left) and panels for different probability threshold. E.g. the top right panel gives the year (x-axis) where exponential emission reduction at different rates (lines) needs to be initiated to limit warming below a given threshold (y-axis) with a probability of 67%. Increasing the required probability tightens the constraint.*

| | $\beta$ | 0.5 | 0.67 | 0.9 | 0.95 | noise-free |
|---|---|---|---|---|---|---|
| **Scenario** | Threshold | | | | | |
| **r = 0.1** | 1.5 K | 2028 | 2024 | 2016 | – | 2027 |
| | 2.0 K | 2046 | 2042 | 2033 | 2028 | 2045 |
| **r = 0.05** | 1.5 K | 2019 | – | – | – | 2018 |
| | 2.0 K | 2038 | 2033 | 2024 | 2020 | 2037 |
| **r = 0.02** | 1.5 K | – | – | – | – | – |
| | 2.0 K | 2022 | 2017 | – | – | 2020 |

*Table RC1: PNR with exponential mitigation at different rates r*

probabilities (top right and bottom panels). Comparing Fig. 2A of Stocker with the top left panel we find that our results are more optimistic than Stocker's, allowing for smaller reduction rates to reach the same target. Our model is more complex than Stocker's, and considering the good reconstruction of relevant RCP scenarios (Figure 4), we have confidence in our results. Under exponential mitigation, the PNR is substantially earlier (Table RC1) when using a value for the exponential reduction rate r that is equal to m1. A problem with exponential pathways is that emissions never reach exactly zero and can still be non-negligible by 2100, e.g. when starting reduction in 2038 at r=0.05 emissions in 2100 still reach 0.56 GtC/yr and 0.26 GtC/yr when starting in 2025. This is difficult to bring into agreement with the "net zero emissions" target of the Paris Agreement. We therefore choose to continue to use the mitigation pathways as defined in the paper.

Changes in Manuscript:
- No changes
- In section 2.3, line 179, we will replace "rate g due" by "rate g = 0.01 due".
- No changes

2. **Comment from referee:**
Uncertainty is only substantively addressed in the text of the appendix. As this is a short text, I suggest to incorporate the appendix into the main text and amplify it. Regarding uncertainty, a general caveat would be useful in the abstract and the conclusion. Otherwise, the stated years of PNR are somewhat misleading.

*Author's response: Uncertainty is an essential part of this work. We assume that the spread in the CMIP5 ensemble captures all kinds of uncertainties, including parameter uncertainties (for example, in climate sensitivity). To this distribution we fit our stochastic model, accounting for all variations between the climate models. Nevertheless, an additional sensitivity study is certainly useful and was performed. We thank the referee for the suggestion and will move the appendix to a subsection at the end of the results section.*

Changes in Manuscript:
At the end of the abstract, we will add the following sentence: "Sensitivity studies show that the PNR is robust with uncertainties of at most a few years.". Table 8 will be adapted visually to allow for easier understanding, the appendix modified appropriately and moved to the end of section 3 (Results).

3. **Comment from referee:**
A constant factor A in the forcing (8b) is used to optimize the agreement with CMIP5. The size of this factor is quite large (1.48). For nalpha_CO2 (in 8b) the correct value is taken (see Tab 3 - however inconsistent parameter notation - only nalpha there!). The authors justify the factor A with the existence of non-CO2 GHG drivers in the CMIP5 results (RCP scenarios), but the effects of these drivers have a time evolution and characteristic time scales that are very different from the primary driver CO2. So I don't quite understand how is it possible to achieve a better match with CMIP5 by using a simple scaling of (8b).

*Author's response:*
*The factor A captures all processes that are not represented in our (simple) model. This includes non-CO2 drivers as well as non-fossil CO2 drivers. In addition, our carbon model and our temperature model come from different model ensembles that are here joined together, and A is a matching factor. Thirdly, as discussed in the paper, the used carbon model is pulse-size-independent, which is a simplification that underestimates concentrations at high emissions. The factor A scales up the forcing from the unrealistically low concentrations to still give the required high radiative forcing.*

Changes in Manuscript:
In line 143, we will replace "and non-CO2 GHG emissions." by "and non-CO2 GHG emissions, as well as matching the carbon and temperature models estimated from different model ensembles) together."

4. **Comment from referee:**
It is not clear, why in (11) both mitigation and abatement are used. Also, there is a conflict of parameters ($a_0$ in 6 and 11). Is 11b, i.e. $a(t)$, really needed and relevant in this paper? I see no discussion in the text or the figures relating to the difference of $m(t)$ and $a(t)$ pathways. In fact, inspecting (11c) I can see no benefit why one would consider both mitigation and abatement. Both have the same linear time dependence, even the same rate. Therefore, the difference seems to lien in the quadratic (positive) contribution $a(t)*m(t)$ to the emission factor, essentially $(m1^2)(t-t_s)^2$, presumably a rather small contribution. Therefore, for simplicity, I suggest that you would eliminate $a(t)$ altogether, which would also remove the parameter conflict of $a_0$.

*Author's response:*
*We thank the referee for noticing the parameter conflict which will be resolved by renaming the coefficients in (6) from $a_0, a_1, a_2, a_3$ to $\mu_0, \mu_1, \mu_2, \mu_3$.*
*We think like many others that there are several important dimensions to climate policy. This includes the substitution of fossil fuel by renewable energies (mitigation) as well as directly reducing the CO2 output via sequestration mechanisms (abatement). We consider it important to include both these dimensions (as a third dimension one might point to negative emissions which we briefly cover as well). It is true that the abatement pathway is chosen very similar to mitigation, both because of simplicity and due to a lack of better estimates. Many now believe that some form of abatement will be necessary, for example to deal with the problem of "stranded assets". Neglecting abatement would clearly require much higher mitigation rates to reach the same targets. For these reasons we decided to include both abatement and mitigation into our modelling framework. Note also that the quadratic term is not necessarily small, for $m1 = 0.02$, $m0 = 0.14$ it reaches >40% of the linear term after 40 years, which slows the decay to zero.*

Changes in Manuscript:
The coefficients in (6), $a_0, a_i$ will be renamed $\mu_0, \mu_i$.

5. **Comment from referee:**
   Further to the emission pathway described in 11c, I note that E_neg is included.
   However, it is not clear from the text, how Fig. 3 is constructed. From the rather short
   caption I surmise that this is taken from Rogelj et al., and then just prescribed here.
   This must be stated in section 2.3 more clearly.

   *Author's response:*
   *The referee is correct that Fig. 3 is constructed from scenarios simply taken from Rogelj et al, as is discussed in the final paragraph of the Methods section.*

   Changes in Manuscript:
   Considering the response to comment 6, resulting in the removal of Fig. 3, no changes will be performed.

6. **Comment from referee:**
   You seem to consider only the strong negative emission of Fig. 3 for the calculation
   of PNR in Tab. 6. As this strong case appears nearly exponential in nature, I would
   suggest that you simply approximate the Rogelj negative emissions by an exponential
   and a starting time, and give it explicitly in eq 11 with its associated rate. This would
   eliminate Fig. 3, be more transparent for the reader and actually more consistent with
   the simple scenario approach that you chose in eq 11.

   *Author's response:*
   *The referee is correct that only results for the strong pathway are presented, so for clarity only it will now be mentioned. We thank the referee for the excellent suggestion to approximate the negative emissions by an exponential. It turns out that the fit is very good.*

   Changes in Manuscript:
   In the first paragraph of section 2.3 the sentence "in addition, negative … concentration." will be replaced by: "In addition, negative emission technologies may be employed. They cause a direct reduction in atmospheric $CO_2$ concentration and are here modelled as an exponential $E\_neg(t) = E_{neg,\infty} * (1 - \exp(r *time))$." A footnote is added to "exponential" in this sentence: "For long timescales, these (after a transient) constant negative emissions may not be realistic. However, we are interested in timescales until 2100."

   The final paragraph of section 2.4 ("Since it is now … (red) pathway.") will be removed.
   Figure 3 will be  removed (in this response we continue to refer to Figures by their label as in the manuscript).
   As the final paragraph of section 2.3 the following will be added:
   "From these scenarios we obtain a family of negative emission scenarios out of which we pick a pathway with strong negative emissions. It is very well approximated by setting $E_{neg,\infty} = 4.21$ and $r=-0.0283$."

7. **Comment from referee:**
   In order to construct ensembles, the mitigation rate $m_1$ is drawn from a Beta distribution.
   It would be helpful for the reader to have an explanation why this distribution is
   chosen and what difference a simple uniform or normal distribution would make.

   *Author's response:*
   *The Beta distribution is chosen for purely practical reasons to get a better coverage of emission scenarios. m0 is drawn from a uniform distribution [0,0.7], so when drawing m1 from e.g. a uniform distribution, many of the m0, m1 pairs would result in a very quick mitigation, resulting in an under-sampling of scenarios with high cumulative emissions. The Beta distribution has the advantage that it is both bounded and (with these parameter values) highly skewed towards small m1, so that the scenario sample is more uniform in terms of cumulative emissions. The choice of distribution has no consequences on the results.*

   Changes in Manuscript:
   In line 245, we will add the following sentence after "latest in 2080." : "The Beta distribution is chosen for practical reasons to a sample of (m0, m1) pairs. As m0 is drawn from a uniform distribution, doing likewise for m1 would result in many pathways with very quick mitigation and low cumulative emissions. Choosing a Beta distribution for m1 makes draws of small m1 much more likely and leading to a better sampling of high cumulative emission scenarios. The choice of distribution has no consequences on the results."

8. Comment from referee:
   *Some noise is added to the model as stated on line 167ff. It seems of only minor relevance for the results (see Tab 5 and 6 - PNR changes only by about 1 year compared to the 50%-probability case). I wonder then why the addition of noise should be necessary at all. I cannot see any new insight from this. If you retain the noise, a more detailed description would be necessary. In particular, the noise should be evident in eqs 10a and 10b as additional terms.*

   *Author's response:*
   *We would like to point the referee to the opening paragraph of this response. Our stochastic state space model consists of four carbon and three temperature boxes, as shown in Table 2. The noise is in several of the carbon and temperature boxes, with $W_t$ denoting the Wiener process. The boxes are simple added (eqs 10a, 10b) to obtain the total, so no additional noise terms are required in this summation. The introduction of additive and multiplicative noise is central to this paper, allowing to get probability distributions (Figure 4). The referee is right to point out the similar values for PNR (Table 5 and 6) for the "noise-free" and 50%-probability case, which is because the deterministic model (setting the noise terms to zero) is very similar to the $50^{th}$ percentile of the distribution (as can be seen in Figure 5). However, the temperature distributions are in fact not symmetric (Figure 4), so (this being a crucial result) the PNR changes substantially when requiring higher safety probabilities \beta (Tables 5 and 6) – in practice, it is likely preferable to have a probability higher than 50% (IPCC works with 67%).*

   Changes in Manuscript:
   We thank the referee for pointing this out and will do our best to clarify the introduction on this point.
   The caption of Table 2 will be changed to the following: "Stochastic State Space Model. Carbon model on the left, temperature model on the right. Wt denotes the Wiener process".
   In line 63, we will replace "stochastic model is then" by "stochastic model – representing all kinds of uncertainties in the climate model ensemble – is then".
   In line 61f, we will replace "stochastic model" by "stochastic state-space model".

9. Comment from referee:
   Table 5, 6, and 7 could be presented in a more effective way. Table 7 is trivial
   (just the difference Tab6 - Tab5) and could therefore be omitted. I further suggest to
   combine Tables 5 and 6 into one table. Each probability column should then contain
   two subcolumns, one without E_neg the other one with E_neg. The small difference
   caused by E_neg makes would then be directly visible.

   *Author's response:*
   *These suggestions are very welcome and the tables will be formatted as suggested.*

   Changes in Manuscript:
   Table 7 will be omitted. Table 6 will be combined into Table 5 by splitting the probability columns into sub-columns, for the case with/without negative emissions.

10. Comment from referee:
    In the appendix and in Tab. 8 some parameters (\gamma_0, r_\gamma) are listed
    without explanation. Where do they come from? Are they needed in this paper?

    *Author's response:*
    *These are parameters connected to related research not included in the final paper. They will be removed.*

    Changes in Manuscript:
    Mentions and discussions of \gamma_0, r_\gamma will be removed from the appendix.

11. Comment from referee:
    Line 374: please spell IPCC correctly. It is an edited document and that information
    is missing, as well as the total page number.

    *Author's response:*
    *We thank the referee for this remark and will correct the formatting.*

    Changes in Manuscript: The reference will be formatted correctly.

12. Comment from referee:
    Figure 2: Put the 10ˆ3 factor into the label unit (1000 ppm).

    *Author's response:*
    *The formatting will be adapted as suggested.*

    Changes in Manuscript:
    The factor of 10^3 in Figure 2, top right panel will be included in the unit label (1000 ppm).

13. Comment from referee:
Figure 2 and line 147. The discrepancy with the CMIP5 CO2 concentrations for RCP8.5 is quite worrying. This would imply that cumulative emissions will be way off, as well. The discrepancy for the forcing is removed by introducing the factor A, but what about CO2(t) and cumE(t)?? This must be addressed in a more convincing way.

*Author's response:*
*Our model has indeed substantial discrepancies in CO2 concentrations for high-emission scenarios such as RCP8.5. The reason for this is the use of a pulse-size-independent carbon response function (essentially meaning that carbon sinks operate at the same efficiency independent of CO2 concentration, temperature, and reservoir sizes). This is introduced in section 2.1 (line 125-139) and discussed in section 4 (lines 310ff). This is indeed a problem for the CO2 concentration, but, as seen in Figure 2, not for radiative forcing or temperature due the factor A (see also comment 3). We are not interested in the intermediate variable CO2(t), and compute cumE(t) directly from the emissions, so this has no substantial effect on our results.*

Changes in Manuscript:
In line 148, we will replace "natural sinks saturate." with "natural sinks saturate, which is a process the pulse-size-independent carbon response function cannot adequately capture."

14. Comment from referee:
Figure 6: Caption should be amplified by elaborating on the "different policies". You could add, e.g.: "... as described by m in eq 11, the rate of mitigation increase per year."

*Author's response:*
*We thank the referee for his suggestion and will incorporate it.*

Changes in Manuscript:
The caption of Figure 6 will be adapted. "different policies, without … negative emissions." is replaced by "different policies as described by in eq 11 with different choices for m1, the rate of mitigation increase per year. Top and bottom panels show the cases without and with strong negative emissions, respectively."

15. Comment from referee:
Figure 7: y-axis labels not complete.

*Author's response:*
*We thank you for the remark and have corrected the labels.*

Changes in Manuscript:
The y-axis labels of Figure 7 will be formatted correctly.

---

## Author Comment (AC3) · 30 May 2018

As suggested by the editor, the title of the paper will be changed into:

The Point of No Return for climate action: effects of climate uncertainty and risk tolerance.